# Assignment of protonated *R*-homocitrate in extracted FeMo-cofactor of nitrogenase via vibrational circular dichroism spectroscopy

Lan Deng[1,5], Hongxin Wang [2,5], Christie H. Dapper[3], William E. Newton[3], Sergey Shilov[4], Shunlin Wang[4], Stephen P. Cramer[2✉] & Zhao-Hui Zhou [1✉]

Protonation of FeMo-cofactor (FeMo-co) is important for the process of substrate hydrogenation. Its structure has been clarified as $\Delta$-Mo*Fe$_7$S$_9$C(*R*-homocit*)(cys)(Hhis) after the efforts of nearly 30 years, but it remains controversial whether FeMo-co is protonated or deprotonated with chelated $\equiv$C $-$ O(H) homocitrate. We have used protonated molybdenum(V) lactate **1** and its enantiomer as model compounds for *R*-homocitrate in FeMo-co of nitrogenase. Vibrational circular dichroism (VCD) spectrum of **1** at 1051 cm$^{-1}$ is attributed to $\equiv$C $-$ O$_H$ vibration, and molybdenum(VI) *R*-lactate at 1086 cm$^{-1}$ is assigned as $\equiv$C $-$ O$_{\alpha\text{-alkoxy}}$ vibration. These vibrations set up labels for the protonation state of coordinated $\alpha$-hydroxycarboxylates. The characteristic VCD band of NMF-extracted FeMo-co is assigned to $\nu$(C $-$ O$_H$), which is based on the comparison of molybdenum(VI) *R*-homocitrate. Density functional theory calculations are consistent with these assignments. To the best of our knowledge, this is the first time that protonated *R*-homocitrate in FeMo-co is confirmed by VCD spectra.

[1] State Key Laboratory of Physical Chemistry of Solid Surfaces and Department of Chemistry, College of Chemistry and Chemical Engineering, Xiamen University, 361005 Xiamen, China. [2] SETI Institute, Mountain View, CA 94043, USA. [3] Department of Biochemistry, Virginia Polytechnic Institute and State University, Blacksburg, VA 24061, USA. [4] Bruker Optics, 19 Fortune Dr., Billerica, MA 01821, USA. [5]These authors contributed equally: Lan Deng, Hongxin Wang. ✉email: scramer@seti.org; zhzhou@xmu.edu.cn

Nitrogenase, as a catalytic enzyme that reduces atmospheric dinitrogen to bioavailable ammonia in nature, has attracted widespread attentions from researchers[1–5]. Molybdenum nitrogenase consists of FeMo- and Fe-proteins, where FeMo-cofactor (FeMo-co) in FeMo-protein is the active site of substrate binding and reduction. Through long-term exploration, the structure of FeMo-co has been clarified as Mo*Fe$_7$S$_9$C($R$-homocit*)(cys)(Hhis)[6–10] (H$_4$homocit = homocitric acid, Hcys = cysteine, Hhis = histidine), and the absolute configuration of molybdenum is assigned as $\Delta$[11]. Homocitrate has been suggested to chelate with metal molybdenum via the oxygen atoms of $\alpha$-alkoxy and $\alpha$-carboxy groups[12]. Nevertheless, besides the metal oxidation states of FeMo-co are ambiguous[13–17], the precise local structure of chelated homocitrate is controversial. With a computational study for a proposed protonated state of FeMo-co[18], structural comparisons of oxidomolybdenum(IV) complexes with the local structures of FeMo-cofactors in Protein Data Bank (PDB) gave an indirect evidence for the protonation of $\alpha$-alkoxy group in $R$-homocitrate[19]. Comparisons of infrared spectroscopy (IR) between FeMo-co and a series of model compounds provided a direct evidence for the two possibilities of $\alpha$-alkoxy and/or $\alpha$-hydroxy groups in $R$-homocitrate[20]. Moreover, quantum mechanics/molecular mechanics calculations recommended protonation of $\alpha$-alkoxy group in FeMo-co[21–24]. But ≡C–O(H) coordination with molybdenum of chelated $R$-homocitrate is still suspicious in view of protons are almost always invisible in limited resolutions of crystal structures, and severe lack of experimental evidence in addition to computational researches[25,26]. The protonation step confers a certain degree of lability to the homocitrate ligation to the cofactor[18,27–29], and thus allows structural flexibility that is important for the mechanism of N$_2$ reduction. Recent capture of N$_2$ species in FeMo-protein confirmed proton translocation and facilitate a serial of hydrogenation like $\alpha$-hydroxy group[30].

Most reported homocitrate and its homologues bind to molybdenum via $\alpha$-alkoxy, $\alpha$-carboxy, and/or $\beta$-carboxy groups[11,31–41]. Several molybdenum $\alpha$-hydroxycarboxylates were isolated coordinating via $\alpha$-hydroxy (protonated) and $\alpha$-carboxy groups[31,42,43]. Less concern is on molybdenum complexes with $\alpha$-hydroxycarboxylate and imidazole-like ligand simultaneously. Here mixed-ligand molybdenum(V) compounds with lactates/glycolate and 1,2,4-triazole [$\Delta$/$\Lambda$-Mo$^*_2$O$_2$($\mu_2$-S)($\mu_2$-O)($R$-Hlact*)$_2$(trz)$_2$(trz)]·½H$_2$O (**1**, H$_2$lact = lactic acid, trz = 1,2,4-triazole), [$\Lambda$/$\Delta$-Mo$^*_2$O$_2$($\mu_2$-S)($\mu_2$-O)($S$-Hlact*)$_2$(trz)$_2$(trz)] (**2**), [Mo$_2$O$_2$($\mu_2$-S)($\mu_2$-O)(Hglyc)$_2$(trz)$_2$(H$_2$O)] (**3**, H$_2$glyc = glycolic acid) obtained in reduced media are selected to imitate the local coordination environment of homocitrate and imidazole in

FeMo-co. The structures and vibrational circular dichroism (VCD) spectra of **1**, **2** and NMF-extracted FeMo-co are compared with those of molybdenum $R$-, $S$-lactates and $R$-homocitrate, as well as the other substituted ≡C–O model compounds. VCD vibration of coordinated ≡C–OH in $R$-homocitrate of extracted FeMo-co is assigned by comparing with chiral molybdenum $\alpha$-hydroxycarboxylates. DFT calculation is consistent with the allocation of VCD spectra.

## Results and discussion

**Structural analyses of 1–3.** Syntheses and possible transformations of dinuclear oxidomolybdenum(V) triazole lactates and glycolates **1–3** are outlined in Supplementary Fig. 1. A series of $\alpha$-hydroxycarboxylic acids have also been tried for the syntheses like glycolate, lactate, malate, citrate and homocitrate, and amino acids such as histidine or cysteine for histidine residue. But most of the obtained Mo(IV/V/VI) complexes are only with $\alpha$-alkoxy coordination. The main factors limiting the syntheses of mixed-ligand complexes were metal valence and pH value. The occurrence of protonation is better under acidic conditions with multi-N coordination sites of 1,2,4-triazole. Complete crystallographic data for **1–3** are summarized in Supplementary Data 1, 3 and Supplementary Table 1. Detailed crystal structures analysis can be seen in Supplementary Figs. 2, 3. Some intermolecular hydrogen bonds of **1–3** are listed in Supplementary Tables 2, 4, respectively. 2D layered diagram of the molecular structures, selected bond distances and angles are shown in Supplementary Figs. 4, 6, Tables 5, 7, respectively. It is noteworthy that there is a small channel existed in the 3D structure of **2** viewed along $a$ axis as shown in Supplementary Fig. 7, while the holes of **1** in the same direction are center-occupied by a free water molecule in Supplementary Fig. 4. Further adsorption tests on **1** and **2** are conducted for some common gases. The results allude they have good adsorptions for O$_2$ and CO$_2$, while no effect for N$_2$, CH$_4$, or H$_2$ in Supplementary Figs. 8, 9 and Tables 8, 9.

**Chiral centers and structural comparisons.** The local coordination modes in **1–3** are similar to those of homocitrate and histidine residue in FeMo-co as shown in Fig. 1a[8–10], where lactate or glycolate imitates homocitrate, and 1,2,4-triazole imitates imidazole residue of histidine respectively. Although there is a large gap between the model compounds and the cofactor of Mo-nitrogenase, the protonated $\alpha$-hydroxy groups in chelated $\alpha$-hydroxycarboxylates should be suitable for the local study of FeMo-co, which possess two chiral centers. Namely chiral metal center Mo in $\Delta$-configuration and chelated homocitrate in $R$

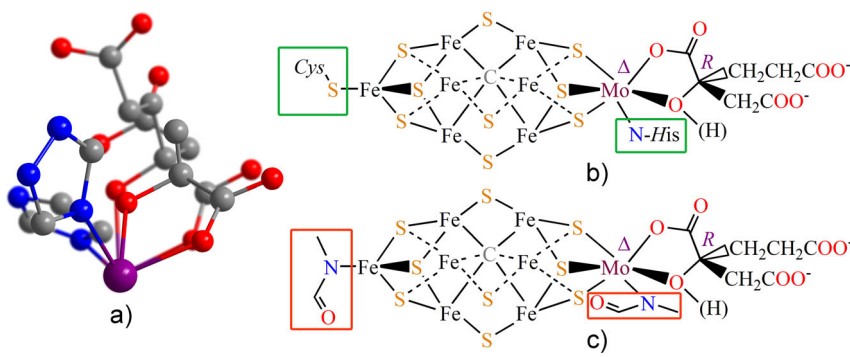

**Fig. 1 Chiral centers and structural comparisons of FeMo-co and model compounds. a** Comparison of the partial Mo-lactate-triazole in **1** (front, opaque) with Mo-homocitrate-histidine coordinated unit in Mo-nitrogenase (behind, at a transparency of 30%)[10]. Color: molybdenum, violet; oxygen, red; nitrogen, blue; carbon, gray. **b** Double chiral centers of FeMo-co in Mo-nitrogenase. **c** Suggested structure for the extracted FeMo-co with $N$-methylformamide (NMF), where amino acids in green boxes of **b** have been replaced by NMF in red boxes of **c**.

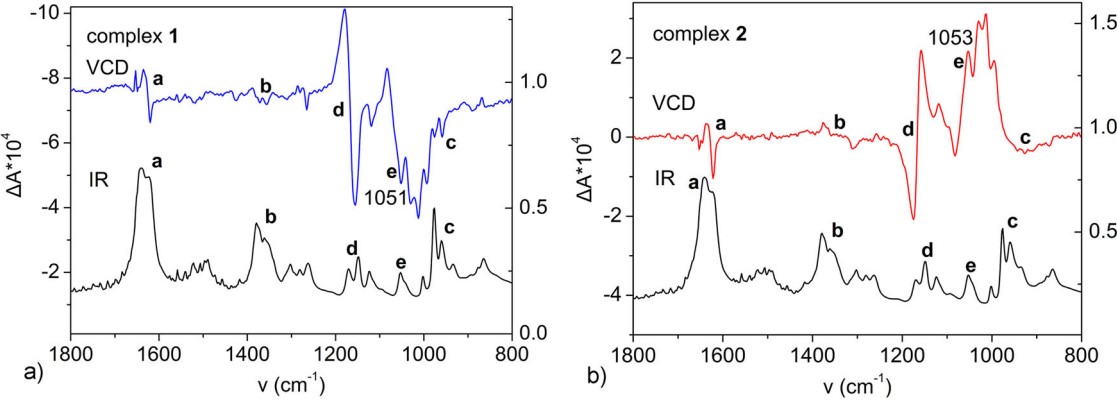

**Fig. 2 VCD and IR spectra of 1–2. a** VCD and IR spectra of [Δ/Λ-Mo*$_2$O$_2$(μ$_2$-S)(μ$_2$-O)(R-Hlact*)$_2$(trz)$_2$(trz)]•½H$_2$O (**1**) and **b** [Λ/Δ-Mo*$_2$O$_2$(μ$_2$-S)(μ$_2$-O)(S-Hlact*)$_2$(trz)$_2$(trz)] (**2**) using KBr pellets.

configuration as shown in Fig. 1b, or the suggested NMF-extracted structure of FeMo-co in Fig. 1c. Due to the influence of the chirality of amino acid residues in MoFe protein, VCD is only used for extracted FeMo-co.

**Analyses of vibrational circular dichroism spectroscopy.** Spectra of vibrational circular dichroism are a powerful tool to investigate chiral molecules in the infrared region, which use different absorptions of left- and right-polarized infrared lights[44–47]. The obtained VCD signals of **1** and **2** were mirror images in the regions of 1800–800 cm$^{-1}$ as shown in Fig. 2. This corresponds to the absolute configurations of Δ/Λ-R and Λ/Δ-S molybdenum lactate with chiral Mo(V) and α-carbon of lactate, respectively. IR spectra obtained are listed below for comparisons. From the VCD spectra, we can see that **1** and **2** show obvious positive and negative Cotton effects. That is, the enantiomers of chiral molybdenum lactates have their chiral characteristics. The chirality of R or S-lactates couples strongly with the chiral molybdenum centers in dinuclear oxidomolybdenum complexes, respectively. Typical coupled peaks around 1652 and 1377 cm$^{-1}$ in peak numbers *a* and *b* are assigned to the asymmetric and symmetric stretching vibrations of CO$_2^-$, respectively[48,49]. The signs of CO$_2^-$ peaks were opposite to each other for both Δ/Λ-R-lactate: +/− and Λ/Δ-S-lactate: −/+. The vibrational bands above 1800 cm$^{-1}$ are assigned to C–H, N–H, and O–H stretching modes[49]. The peaks (numbered as *c*) in the region of 967–977 cm$^{-1}$ indicate the existence of Mo=O bonds, which are consistent with the values observed in IR spectra. The bands in the range of 1120–1170 cm$^{-1}$ for peak numbers *d* are associated with C–N vibrations of triazole compared with IR spectra of free 1,2,4-triazole in Supplementary Fig. 23[50,51].

The C–O stretching vibrations in alcohols produce bands in the region of 1100–1000 cm$^{-1}$. In our previous report, IR spectra of some alcohols, α-hydroxycarboxylic acids have been compared with Spectral Database for Organic Compounds (SDBS), where peak 1047 cm$^{-1}$ is assigned to C–O$_H$ vibration for lactic acid in Supplementary Fig. 24[20]. Here strong VCD peaks at 1051 in **1** and 1053 cm$^{-1}$ in **2** are assigned to the protonated C–OH stretching vibrations numbered as *e* in Fig. 2, respectively. These are in agreement with those obtained from IR spectra simultaneously. Compared to peaks *e*, peaks *a* ~ *d* are not very strong. This is because these characteristic vibrations are located far away from the chiral carbon, and triazole is not chiral. Likewise, VCD and IR spectra of deprotonated {Na$_2$[Δ-Mo*O$_2$(R-lact*)$_2$]}$_3$ · 13H$_2$O, {Na$_2$[Λ-Mo*O$_2$(S-lact*)$_2$]}$_3$ · 13H$_2$O[38] are shown in Supplementary Fig. 10. By comparing with VCD

spectra of deprotonated {Na$_2$[Δ-Mo*O$_2$(R-lact*)$_2$]}$_3$ · 13H$_2$O with Δ/Λ-R configuration in protonated dimer **1** in Fig. 3, we can draw the conclusion that protonated C–OH stretching vibrational frequencies are generally lower than those of deprotonated C–O stretching vibrations, where strong vibrations at 1086 cm$^{-1}$ for {Na$_2$[Δ-Mo*O$_2$(R-lact*)$_2$]}$_3$ · 13H$_2$O is assigned to the deprotonated C–O$_{\alpha\text{-alkoxy}}$ group. More diagnostic peaks are shown in Supplementary Fig. 11. In the other words, ν(C–OH) will shift from high wave number to low wave number, when R-lactate undergoes protonated coordination. This is also consistent with the auxiliary infrared results. The protonation weakens the strength of C–O bond, resulting in the red-shift of vibrational frequency. Combined with C–O(H) bond distances discussed in Supplementary Tables 10 and 11, the conclusion of red-shift is also in compliance with Hooke's law. The same result also appears in Λ/Δ-molybdenum S-lactate **2** and {Na$_2$[Λ-Mo*O$_2$(S-lact*)$_2$]}$_3$ · 13H$_2$O[38], which can be seen in Supplementary Fig. 12.

To identify the protonation state of homocitrate in FeMo-co, we have measured the VCD spectrum of extracted FeMo-co Δ-Mo*Fe$_7$S$_9$C[R-(H)homocit*](NMF)$_2$ (**20**) purified from nitrogenase in *Azotobacter vinelandii*, where coordination sites of histidine and cysteine residues have been substituted by N-methylformamides. The VCD spectrum of FeMo-co is compared with that of the synthetic deprotonated molybdenum(VI) R-homocitrate K$_5$[Λ,Λ,Λ,Λ-Mo*$_4$O$_{11}$(R-Hhomocit*)$_2$]Cl·5H$_2$O (**21**)[33] in Fig. 4, where the film spectrum of extracted FeMo-co **20** exhibits absorption peaks at 1657, 1622, 1608, 1527, 1479, 1429, 1383, 1327, 1271, 1189, 1153, 1068, and 957 cm$^{-1}$, respectively. Qualitative assignments of **20** and **21** are given in Supplementary Table 12. The vibrations of the following groups CO$_2^-$, C–C, C–O(H), Mo=O, and NMF are identified. IR spectrum of pure NMF is listed in Supplementary Fig. 25 for comparison. The C–O stretching frequencies of molybdenum R-homocitrate **21** can be served as references for C–OH vibrations in FeMo-co **20**, and also those from the previous assignments of infrared spectra[20]. Here VCD spectra are much more sensitive only for the local environments around the chiral organic component of R-homocitrate and metal center of molybdenum atom than those from infrared spectra. We can assign the peak of 1068 cm$^{-1}$ as C–OH vibrational frequency for FeMo-co, which are similar to the assignments of C–OH vibrations in **1**, **2**.

VCD peaks around 1657 cm$^{-1}$, 1622 cm$^{-1}$ in **20**, and 1616 cm$^{-1}$ in **21** are assigned to the asymmetric vibration ν$_{as}$(CO$_2^-$) for R-homocitrate, respectively, while 1705 cm$^{-1}$ for free γ-carboxylic acidic group in **21**. The peaks at 1479 and 1383 cm$^{-1}$ for **20** should belong to the symmetric carboxy vibrations. Those at 1608 and 1327 cm$^{-1}$ in FeMo-co **20** are assigned to the asymmetric and

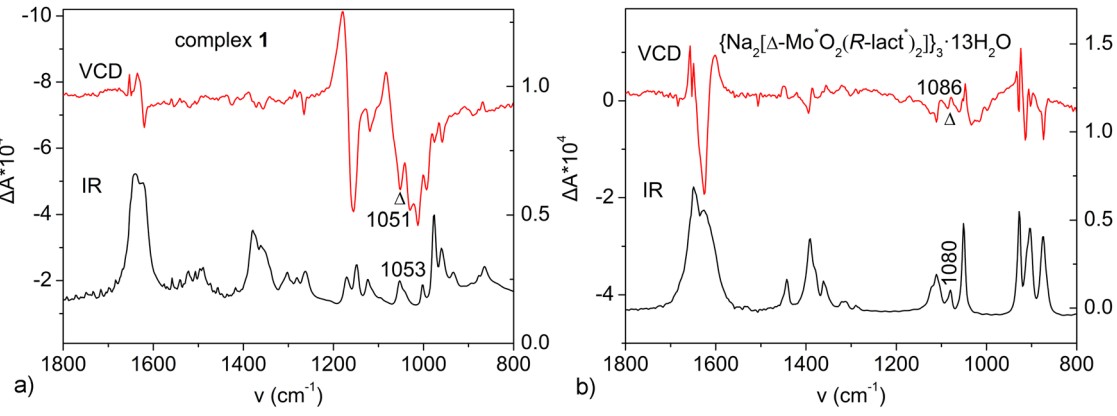

**Fig. 3 VCD and IR spectra of 1 and {Na₂[Δ-Mo*O₂(R-lact*)₂]}₃ • 13H₂O. a** Comparison of VCD and IR spectra of [Δ/Λ-Mo*₂O₂(μ₂-S)(μ₂-O)(R-Hlact*)₂(trz)₂(trz)]•½H₂O (**1**) and **b** {Na₂[Δ-Mo*O₂(R-lact*)₂]}₃ • 13H₂O.

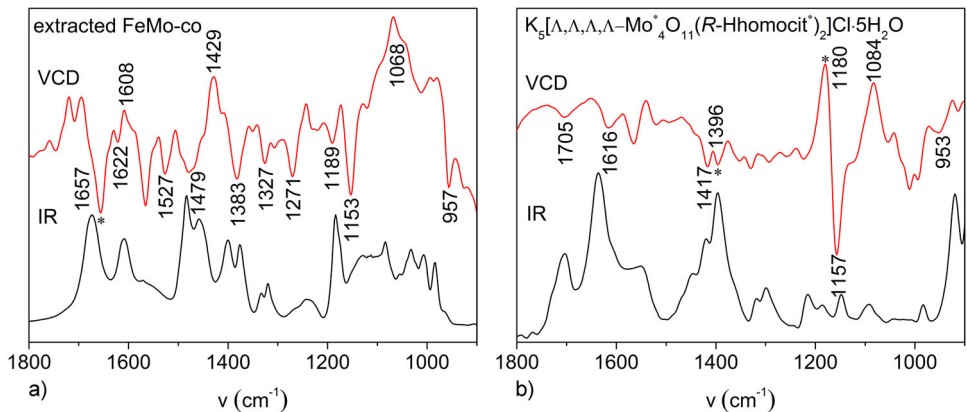

**Fig. 4 VCD and IR spectra of 20–21. a** VCD and IR spectra of extracted FeMo-co (**20**) and **b** molybdenum(VI) R-homocitrate (**21**) in the regions of 1800–900 cm⁻¹.

symmetric vibrations ν(−NHCO) for NMF absorptions, while the peak at 1527 cm⁻¹ should belong to N − H vibration of *N*-methylformamide, which disappeared in **21** referred to IR spectra of free NMF (Supplementary Fig. 25)[32,52]. The peaks at 1189 cm⁻¹ and 1153 cm⁻¹ for **20** are assigned to C–C vibrations, and the peak at 953 cm⁻¹ in **21** is assigned to Mo=O vibration.

Most notably, the peak at 1068 cm⁻¹ in the VCD spectrum of extracted FeMo-co **20** is assigned to C–OH vibration as mentioned above. Similar absorptions at 1051 cm⁻¹ in **1**, 1053 cm⁻¹ in **2** are also observed for C–OH vibrations respectively. While the peak at 1084 cm⁻¹ in **21** is assigned to the deprotonated C–O vibration, which is in similar tendency for C–O vibration of 1086 cm⁻¹ in {Na₂[Δ-Mo*O₂(R-lact*)₂]}₃ · 13H₂O[38]. The ≡C–O(H) peaks in compounds Na₃(Hhomocit)·H₂O[53], K₂[MoᵛᴵO₂(R,S-H₂homocit)₂]·2H₂O[53], and Na₂[Mo₃-SO₃(R,S-lact)₃(im)₃]·10H₂O[19] shown in Supplementary Figs. 26, 28 are used for references. When homocitrate undergoes protonation, C–OH vibration will shift from high wave number to low wave number, which is consistent with the red-shift observed for protonated *R*-lactate described above. Unlike VCD vibrations of molybdenum *R*- or *S*-lactates coordinated with α-alkoxy (1086, 1078 cm⁻¹) or α-hydroxy groups (1051, 1053 cm⁻¹), the difference is smaller between α-alkoxy (1084 cm⁻¹) and α-hydroxy groups (1068 cm⁻¹) for molybdenum *R*-homocitrates. This is attributed to their electron-drawing effects from side chains of β- and γ- carboxy groups in *R*-homocitrate. Therefore, protonated homocitrate in extracted

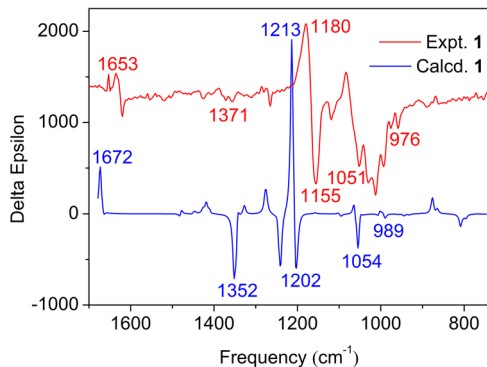

**Fig. 5 Experimental and theoretical VCD spectra of 1.** Comparison of the experimental and theoretical VCD spectra of [Δ/Λ-Mo*₂O₂(μ₂-S)(μ₂-O)(R-Hlact*)₂(trz)₂(trz)]•½H₂O (**1**).

FeMo-co has been evaluated through the comparisons of VCD spectra with molybdenum complexes of chiral protonated and deprotonated *R, S*-lactates and *R*-homocitrates. That is, coordinated *R*-homocitrate in FeMo-co of Mo-nitrogenase should protonate at resting state, as those of N₂-coordinated FeMo-co with free or coordinated α-hydroxy groups[30].

**Density functional theory calculations.** Theoretical calculations are useful to identify the vibrational modes of the chiral molecule

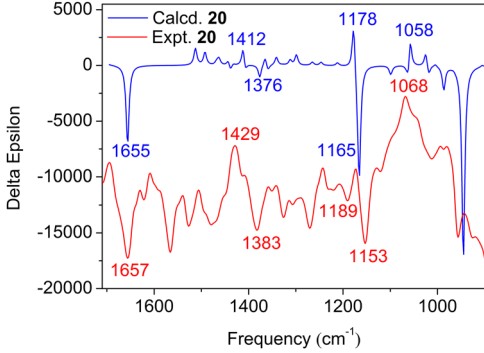

**Fig. 6 Experimental and calculated VCD spectra of 20.** Comparison of the experimental and calculated VCD spectra for NMF-extracted FeMo-co Δ-Mo*Fe₇S₉C[R-(H)homocit*](NMF)₂ (**20**).

for its VCD assignments. The optimized structures of molybdenum *R*-lactate **1** and NMF-extracted FeMo-co **20** have been calculated by density functional theory (DFT) at the B3LYP/def2-SVP level. Figure 5 shows experimental and calculated VCD spectra of **1**. The vibrational modes of the experimental VCD signals agreed well with those calculated. The calculated $\nu_{as}$(C=O) and $\nu_s$(C=O) peaks of **1** appeared at 1653 and 1371 cm⁻¹. This is close to the observed bands with the signs of peaks *a*, *b* in **1**. The calculated signals of N–H and Mo=O vibrations modes agree with the experimental ones. The $\nu$(C–O$_H$) peak that we care about mostly appears in a similar position at 1054 cm⁻¹, where 1051 cm⁻¹ is for the experimental VCD of **1**. This further increases the accuracy of aforementioned assignments. Optimized molecular structure of **1** is shown in Supplementary Fig. 13. Specific peaks for identification can be seen in Supplementary Table 13.

We further studied the experimental and theoretical VCD spectra of extracted FeMo-co **20** synchronously shown in Fig. 6. We can draw a conclusion that the vibrational signals of VCD obtained from experiment are basically consistent with the calculated values. 1058 cm⁻¹ is identified as $\nu$(C–O$_H$) peak. Optimized molecular structure of **20** is shown in Supplementary Fig. 14. More data for peak identification can be seen in Supplementary Table 14.

The upshots of theoretical calculations tell us that protonation will affect the VCD signal of the entire chiral structure. These further confirm the accuracy of the VCD analysis for different protonated model compounds, indicating that coordinated *R*-homocitrate in the extracted FeMo-co is protonated. Therefore, we suggested the protonation of *α*-alkoxy groups in *R*-homocitrate of Mo-nitrogenase.

**Other spectral and data analyses.** IR spectra of **1**–**3** in different regions, solid diffused reflectance UV-Vis spectra are shown in Supplementary Figs. 15, 17, respectively. Solid-state ¹H and ¹³C NMR spectra, thermogravimetric analysis, differential thermal analyses and electron paramagnetic resonance spectra for three complexes are shown in Supplementary Figs. 18, 22, respectively. Theoretical bond valence calculations data can be seen in Supplementary Table 15. Comparisons of Mo–O$_{α\text{-hydroxy}/α\text{-alkoxy}}$, Mo–O$_{α\text{-carboxy}}$, C–O$_{α\text{-hydroxy}/α\text{-alkoxy}}$ distances (Å) for **1**–**3** with different metal-valence molybdenum lactates, glycolates are shown in Supplementary Table 10. The characteristic bond distances (Å) for 49 reported structures of FeMo-cofactors in the RCSB Protein Data Bank are listed in Supplementary Table 11. From the contradistinctive consequences we can see the protonation state of *α*-alkoxy groups have more obvious influences in bond distances than those from oxidation state of metal or ligand types.

**Conclusions.** We describe a study that examines the protonation state of homocitrate in FeMo-co through a comparative spectroscopic analysis of NMF-extracted FeMo-co with chiral molybdenum *α*-hydroxycarboxylates. Protonated dinuclear molybdenum(V) lactates [Δ/Λ-Mo*₂O₂(μ₂-S)(μ₂-O)(*R*-Hlact*)₂(trz)₂(trz)]·½H₂O (**1**), [Λ/Δ-Mo*₂O₂(μ₂-S)(μ₂-O)(*S*-Hlact*)₂(trz)₂(trz)] (**2**), and glycolate [Mo₂O₂(μ₂-S)(μ₂-O)(Hglyc)₂(trz)₂(H₂O)] (**3**) have been used as mimic compounds for the local chelated environment of *R*-homocitrate in FeMo-cofactor. The lactates/glycolates in **1**–**3** chelate to molybdenum (V) atoms with *α*-hydroxy and *α*-carboxy groups as bidentate ligands respectively. The longer distances of Mo–O$_{α\text{-hydroxy}}$ [2.243(9)$_{av}$ Å in **1**, 2.246(5)$_{av}$ Å in **2**, 2.284(7)$_{av}$ Å in **3**] are comparable with that of FeMo-co **20** (Mo–O$_{α\text{-hydroxy}}$ 2.263$_{av}$ Å). Most importantly, the protonation state of *R*-homocitrate in FeMo-co has been suggested based on comparisons of VCD spectra of NMF-extracted FeMo-co with **1**, **2** and molybdenum *R*-homocitrate K₅[Λ,Λ,Λ,Λ-Mo*₄O₁₁(*R*-Hhomocit*)₂]Cl·5H₂O. The VCD specta can be served as a strong evidence for the protonation of *R*-homocitrate in FeMo-co, and DFT calculations have increased credibility of the distributive results. The presence of protonation has implications regarding the nature of protonated FeMo-co redox states as well as for potential substrate reduction mechanisms for hydrogenation. VCD spectroscopy for the protonation state of *R*-homocitrate opens up an exciting direction in investigating the molecular-mechanism of biological nitrogen fixation.

## Methods

**Preparations of [Δ/Λ-Mo*₂O₂(μ₂-S)(μ₂-O)(*R*-Hlact*)₂(trz)₂(trz)]•½H₂O (1).** Na₂MoO₄ · 2H₂O (1.21 g, 5.0 mmol) and excess *R*-lactic acid (1.0 mL, 13.0 mmol) were dissolved in 9.0 mL water. The pH value of the solution was adjusted to 4.0 with the addition of NaOH. The mixture was heated at 60 °C for 24 h and cooled to room temperature. Excess Na₂S₂O₄ (0.871 g, 5.0 mmol) and 1,2,4-triazole (0.691 g, 10.0 mmol) were added and the pH was controlled to 2.0–3.0 with concentrated hydrochloric acid. The mixture was then heated at 80 °C for 24 h and cooled to room temperature. Product **1** was isolated as yellow plates immediately in 40.9% yields (0.682 g) based on molybdenum. Elemental analysis (calc. for C₁₂H₂₀Mo₂-N₉O₉.₅S): C, 21.6; H, 3.0; N, 18.9%. Found: C, 21.4; H, 3.0; N, 18.7%. mp: 474 °C (decompose, molybdenum oxide). IR (KBr, cm⁻¹): $\nu$(OH) 3448$_{br, s}$; $\nu$(C–H) 3127$_{vs}$, 2941$_m$, 2856$_w$, 2346$_w$; $\nu_{as}$(CO₂) 1638$_{vs}$; $\nu_s$(CO₂) 1380$_s$; $\nu$(Mo=O) 977$_s$. UV(H₂O, nm): 323. Solid-state ¹H NMR (400 MHz, ADA, ppm): δ 8.64 (s, 1H), 3.83 (s, 1H), 0.60 (s, 3H). Solid-state ¹³C NMR (400 MHz, 25 °C, ADA, ppm): δ = 180.11 (**C**O₂), 154.38–142.28 (trz), 70.40 (**C**OH), 19.77 (**C**H₃).

**Cell growth and purification of nitrogenase proteins.** The *Av* wild-type strain was grown in the absence of a fixed-nitrogen source in a 24-L fermenter at 30 °C in a modified, liquid Burk medium[54]. All cultures contained 20 μM FeCl₃ and 10 μM Na₂MoO₄ and were grown to a final cell density of 250 Klett units recorded on a Klett–Summerson meter equipped with a number 54 filter. All manipulations of nitrogenase proteins were performed anaerobically using either a Schlenk line or an anaerobic glovebox operating at less than 1 ppm of O₂. After harvesting, cell extracts were prepared by diluting the whole cells with an equal amount of 50 mM Tris pH 8.0 buffer prior to passing through a French pressure cell and a centrifuge at 98,000 × *g* for 90 min. Nitrogenase component proteins were separated by anaerobic Q-Sepharose anion exchange column chromatography using a linear NaCl concentration gradient. *Av*2 was purified to homogeneity by fractionation from a second Q-Sepharose column. *Av*1 was further purified by Sephacryl S-200 gel filtration and phenyl-Sepharose hydrophobic-interaction chromatography[55]. The purified nitrogenase proteins were concentrated individually using an Amicon microfiltration pressure concentrator before buffer exchange to 25 mM HEPES pH 7.5, 100 mM NaCl, 10 mM MgCl₂, and 2 mM Na₂S₂O₄ by dialysis at 4 °C. Purified wild-type *Av*1 had specific activities of 2200 nmol of H₂ (min·mg·protein)⁻¹ at 30 °C, when assayed in the presence of an optimal amount of the purified complementary component protein as described previously[55]. Protein concentrations were determined by the Lowry method.

**Extraction of FeMo-co from *Av*1 and activity assays.** *Av*1 was purified as above through the gel-filtration step, yielding protein with a specific activity of about 1000 nmol of H₂ (min·mg protein)⁻¹ and a Mo content of ~1 g·atom per mol of *Av*1. After dialysis to lower the NaCl concentration, the *Av*1 was loaded onto a DE-52 cellulose column that had been washed with 50 mM Tris pH 7.4 buffer containing 2 mM Na₂S₂O₄. The bound protein was washed with *N,N*-dimethylformamide

containing 50 mM 2,2′- bipyridine, 5 mM phosphate buffer pH 8, with 2 mM $Na_2S_2O_4$, and water (ca. 5% v/v) until the non-cofactor iron was completely eluted. The column was then washed with N-methylformamide (NMF) containing 5 mM phosphate buffer pH 8, with 2 mM $Na_2S_2O_4$, and water (ca. 5% v/v), and FeMo-co was then eluted with NMF that contained 500 mM tetraethylammonium chloride, 5 mM phosphate buffer pH 8, with 2 mM $Na_2S_2O_4$, and water (ca. 5% v/v). The eluted FeMo-co was concentrated approximately 20-fold by distilling off the NMF under vacuum at 40 °C. FeMo-co was assayed by reconstitution of the DJ42 Av strain[56], which has a deletion for the FeMo-co biosynthetic genes nifENX. The FeMo-co used in this study activated a DJ42 crude extract and produced 75 nmol of $H_2$ (min·mg protein)$^{-1}$. The extracted FeMo-co can be used to reactivate apo-nitrogenase, activation of the FeMo-co-deficient MoFe protein by NMF-extracted FeMo-co were determined as document described and will not affect recombination activity[57–59]. The extraction process has nothing to do with the redox state including protonation state of FeMo-co[60]. Moreover, hydrocarbon formation like $CH_4$, $C_2H_4$, $C_2H_6$ by solvent-extracted cofactors proved CO can be reduced by cofactors in the presence of strong chemical reductants without the assistance of corresponding protein scaffolds[61,62]. But protein environment still is a major contributor in view of the activity of FeMo-co[63]. Namely, the catalytic activity of NMF-extracted FeMo-co can maintain in CO and $N_2$ reductions with the chelated mode of R-homocitrate, regarding the recent structures bound with CO and $N_2$ substrates[10,30].

**Physical measurements**. $Na_2MoO_4·2H_2O$, R-lactic acid, 1,2,4-triazole and $Na_2S_2O_4$ were purchased from Sigma. All solvents and chemicals were of commercially analytical grade. pH value was determined by PHB-8 digital pH meter. Elemental analyses (for C, H, and N) were performed with an Vario ELIII elemental analyzer. Spectra of vibrational circular dichroism for **1**, **2** and {$Na_2[\Delta$-$Mo^*O_2(R$-$lact^*)_2]\}_3 · 13H_2O$, {$Na_2[\Lambda$-$Mo^*O_2(S$-$lact^*)_2]\}_3 · 13H_2O$ were recorded on a BioTools Chiral IR-2X spectrometer, where solid samples were mixed in KBr pellets at a mass ratio of 1:100, respectively. The VCD spectra of extracted FeMo-co and $K_5[\Lambda,\Lambda,\Lambda,\Lambda$-$Mo^*_4O_{11}(R$-$Hhomocit^*)_2]Cl·5H_2O$ were measured by Bruker VCD/IRRAS module PMA 37 with MCT detector. To minimize the NMF left in the FeMo-co film, the samples were pumped for more than 8 h before the measurement. More physical measurements, preparations of $[\Lambda/\Delta$-$Mo^*_2O_2(\mu_2$-$S)(\mu_2$-$O)(S$-$Hlact^*)_2(trz)_2(trz)]$ (**2**), $[Mo_2O_2(\mu_2$-$S)(\mu_2$-$O)(Hglyc)_2(trz)_2(H_2O)]$ (**3**), {$Na_2[\Delta$-$Mo^*O_2(R$-$lact^*)_2]\}_3 · 13H_2O$ and $K_5[\Lambda,\Lambda,\Lambda,\Lambda$-$Mo^*_4O_{11}(R$-$Hhomocit^*)_2]Cl·5H_2O$ (**21**), gas adsorption and X-ray crystallography can be seen in the Part I of the Supplementary Methods of Supporting Information.

**Theoretical calculations**. All optimizations for the electronic structures were calculated at B3LYP/def2-SVP level for the two systems. VCD spectra for **1** and **20** were also culcalated under this level. A least-squares approach has been used to determine multiplicative scaling factors for harmonic vibrational frequencies to facilitate comparison with experimentally observed frequencies, under the deriction of Computational Chemistry Comparison and Benchmark DataBase (CCCBDB)[64–67]. The dispersion correction was conducted by Grimme's D3 version with BJ damping function[68]. All quantum calculations were done with Gaussian16a software.

**Reporting summary**. Further information on research design is available in the Nature Research Reporting Summary linked to this article.

## Data availability

All data that support the findings in this study are available within the article and its Supplementary Information and/or from the corresponding authors on reasonable request. The X-ray crystallographic coordinates for structures reported in this Article have been deposited at the Cambridge Crystallographic Data Centre (CCDC), under deposition numbers 1993896–1993898. These data can be obtained free of charge from The Cambridge Crystallographic Data Centre via http://www.ccdc.cam.ac.uk/data_request/cif.

## Code availability

The Gaussian 16 package, Revision A.03 was employed for all the DFT calculations[69]. The software and instructions for its use are available at http://gaussian.com/.

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

## Acknowledgements

We thank the support from the National Natural Science Foundation (21773196) for the generous financial supports and thank Fan Yao from Shiyanjia Lab (www.shiyanjia.com) for the VCD calculations.

## Author contributions

L.D. and H.W. performed the syntheses, spectral characterizations, and structure analyses. C.H.D., W.E.N. extracted FeMo-co. S.S. and S.W. conducted VCD spectroscopic measurements. S.P.C. designed VCD experiment of FeMo-co. L.D. and Z.H.Z. conducted VCD experiments and prepared the manuscript with feedback from the other authors.

## Competing interests

The authors declare no competing interests.
