## [Peer Review File · Communications Chemistry]

Reviewers' comments:

Reviewer #1 (Remarks to the Author):

The paper by Deng et al describes an experimental study about assigning the protonation state of the homocitrate ligand in the FeMo cofactor of nitrogenase using VCD spectroscopy, comparing extracted cofactor to model molybdenum compounds. As the mechanism of nitrogenase is still not well understood and the homocitrate has been implicated in the reaction this is certainly a worthwhile area to study.

The authors have synthesized interesting dinuclear oxothiomolybdenum compounds that mimic somewhat the Mo ligand environment in FeMoco (Mo binds 3 sulfides, homocitrate via the oxygens of an alkoxy group and a carboxylate group and to N in imidazole in FeMoco). Of concern is the high oxidation state (V) in the model compounds compared to FeMoco (III) and the Mo-Mo bond (I'm assuming) in the dinuclear compounds that is less similar to FeMoco. Also it is not clear how relevant the protonation state of extracted FeMoco is to the protein-bound FeMoco making the study less relevant for the enzyme chemistry. The structure of extracted cofactor is not completely known, there has been no crystal structure as far as I am aware. It is obvious that the cysteine and histidine ligands are no longer present, plausibly being replaced by NMF solvent ligands but it is not clear whether there are other structural changes in going from protein-bound cofactor to cofactor NMF solution. The vibrational assignments made are also not entirely convincing. Is there no structure available for compound 13? Since extracted FeMoco is being compared to this compound that seems like a crucial omission. I would also strongly suggest the authors to collaborate with a computational chemist to put these band assignments on firmer ground.

Overall, while the work is interesting, the assignments are not entirely convincing and as this is extracted cofactor (without a completely known structure) means that the study is less relevant for nitrogenase and should rather be published in a more specialized journal.

Other comments:

- The story is not well presented and there are inconsistencies in the text, e.g. Line 199-200: "When homocitrate undergoes deprotonation, C–OH vibration will shift from 1084 to 1068 cm⁻¹", shouldn't this statement be the other way around?
- There are some language issues that should be taken care of.
- Line 30: The abbreviations used in the chemical formula of FeMoco seem inconsistent.
- Line 199-200: I don't think it is fair to say that when homocitrate is deprotonated, it shifts from 1068 cm⁻¹ to 1084 cm⁻¹ as these are two different compounds that are being compared.
- Figure 6: The peaks are labelled with bold numbers (1,2,3,4,5,6,7,8,9,10,11,12,13) which is the same as used when discussing the compounds. It would make the discussion clearer to use different labels.

Reviewer #2 (Remarks to the Author):

In the present work, Zhou and co-workers synthesized and characterized two dinuclear (lactate)Mo(V) oxo/sulfido complexes 1 and 2 and a glycolate complex 3. They measured the VCD spectra of 1, 2, some known Mo(VI)-oxo complexes, and extracted FeMo-cofactor, to analyze the possible protonation on the alpha-hydroxy group of R-homocitrate in the extracted FeMoco. In particular, the C-O/C-OH stretching frequencies were compared among the Mo complexes and extracted FeMoco, and the authors concluded the presence of a proton on the alpha-oxygen atom. The importance of the presence of a proton on the chelating alpha-hydroxy group of R-homocitrate (in the extracted FeMoco) has not been sufficiently discussed. Therefore, the importance of the work in the community remains unclear.

What the authors mainly carried out in this work were synthesis, characterization, and spectroscopic analysis of a few Mo(V) oxo/sulfido complexes, which were unfortunately unable to assign the entitled "protonated R-homocitrate in FeMo-cofactor". Therefore the title is misleading, in my opinion. The observed C-O frequency of 1068 cm⁻¹ for the extracted FeMo-cofactor is in between the C-O/C-OH vibrations of lactate complexes. As such, the authors cannot conclude the C-OH moiety in the FeMoco based on the new compounds. The authors finally proposed the C-OH moiety in the extracted FeMoco, based on the comparison with a known (R-homocitrate)Mo(VI) oxo complex, which exhibits the C-O band at 1084 cm⁻¹. Thus, the conclusive evidence is not from the main body of the present work.

The number of closely relevant Mo compounds exceeds 50 for Mo(V) or Mo(VI) oxo or oxo/sulfido complexes, in which Mo atoms are supported by an N-donor ligand and a chelating carboxylate/alcohol or alkoxide. Therefore, synthesis and characterization of the present Mo complexes do not stand out.

With these reasons, I am sorry to conclude that the present work does not warrant sufficient novelty required for Communications Chemistry.

Additional comments:

- a) Comparison of the C-O/C-OH frequencies of chelating lactate was made by compounds in different oxidation states, Mo(V) for C-OH and Mo(VI) for C-O. The authors need to clarify the effect of oxidation states before attributing the bathochromic shift to protonation/deprotonation behaviors. The Mo-O bond strength varies by the oxidation state, and the Mo-O bond strength should affect the neighboring C-O bond strength.
- b) Along these lines, the oxidation states of "model" compounds are different from the extracted FeMoco. The Mo(V) and Mo(VI) states are irrelevant to the Mo atom of FeMo-cofactor, which has been assigned as the Mo(III) state (in some cases proposed as the Mo(IV) state).

Reviewer #3 (Remarks to the Author):

The manuscript by Deng et al. describes a study that examines the protonation state of the homocitrate ligand of the FeMo-cofactor through a comparative spectroscopic analysis of the extracted FeMo-cofactor of Mo-nitrogenase with four organo-molybdenum compounds. While the available crystal structures of MoFe protein suggest that homocitrate is coordinated to the Mo atom of the FeMo-cofactor through the O atoms of its carboxyl (COO) and alkoxyl (CO) groups, recent theoretical studies implied that the alkoxyl ligand might be protonated into a COH group (C2 atom). Based on these calculations, it has been proposed that this protonation step confers a certain degree of lability to the homocitrate ligation to the cofactor, and thus allows structural flexibility that is important for the mechanism of N₂ reduction. The current study takes advantage of the chirality of the homocitrate COH group (C2 atom), which can be probed by vibrational circular dichroism (VCD) spectroscopy. The authors first characterized two Mo complexes coordinated by the R- and S-enantiomers of protonated-lactates (Hlac) and identified the COH vibrations around the 1050-1060 cm region through the mirrored VCD spectra of the two enantiomers. They then studied two Mo complexes coordinated by R- and S-lactates (lac) and identified their CO- vibrations to be at around 1070-1090 cm. Taken together, these results established that the protonated ligand had a CO(H) vibration at lower wavenumbers compared to its non-protonated counterparts. Based on these results, the authors then studied the NMF-extracted FeMo-cofactor using the same VCD technique and identified a peak at 1068 cm that was consistent with a vibration at 1084 cm and observed in the VCD spectra of an unprotonated R-homocitrate ligand of the Mo compound. This observation led the authors to conclude that the homocitrate of the FeMo-cofactor is ligated to Mo through a protonated COH group.

This study represents the first direct experimental attempt to prove the existence of a protonated ligation of homocitrate to the Mo atom of the FeMo-cofactor, which is of potential relevance to the mechanism of nitrogenase. However, despite its scientific appeal, the manuscript suffers from two major issues. One, the justification for the assignment of the VCD peaks to certain vibrations is unclear. For example, the authors based the crucial assignment of the peak at 1068 cm⁻¹ in the spectrum of FeMo-cofactor on the similarity of this peak to those observed in the spectra of the Mo complexes. The basis for such an assignment is not apparent to the non-experts. In addition, the VCD spectra presented by the authors are rather complex convoluted, and the interpretation of these spectra lacks any validation by results derived from complementary methods, such as DFT simulations and/or isotopic labeling experiments. Two, the relevance of the extracted FeMo-cofactor to its native protein bound counterpart is unclear. Given the fragile nature of the extracted FeMo-cofactor, it is questionable that the cofactor would remain intact after the extensive (8 hr) drying procedure. Moreover, even if the homocitrate ligand of the FeMo-cofactor indeed undergoes protonation, it is possible that this event occurred during the extraction procedure. Concerns along this line are not addressed by experiments or discussed in the manuscript and prevent publication of this manuscript in the current state.

There are also some minor issues that should be addressed by the authors:

1. Page 3, line 27: "their cofactor" should be "their cofactors".
2. The overlay of the two moieties in Figure 2a is hard to visualize. It would be helpful to use different levels of transparency or color schemes for each moiety.
3. The NMR data presented in the manuscript are not necessarily supportive for the conclusion of this study. These data should be moved into the Supplemental Materials.
4. Illustrations of all compounds used in this study should be included in the manuscript.
5. A discussion of the significance of the findings of this work for a better understanding of nitrogenase mechanism should be included in the Conclusion section.
6. Many parts of the manuscript are difficult to read (e.g., line 30-31, line 78-81, and line 140-143, etc.). The manuscript would benefit from a thorough proofreading for improved clarity.

Response to reviewers' comments

Reviewers' comments:

Reviewer #1: The paper by Deng et al describes an experimental study about assigning the protonation state of the homocitrate ligand in the FeMo cofactor of nitrogenase using VCD spectroscopy, comparing extracted cofactor to model molybdenum compounds. As the mechanism of nitrogenase is still not well understood and the homocitrate has been implicated in the reaction this is certainly a worthwhile area to study.

The authors have synthesized interesting dinuclear oxothiomolybdenum compounds that mimic somewhat the Mo ligand environment in FeMoco (Mo binds 3 sulfides, homocitrate via the oxygens of an alkoxy group and a carboxylate group and to N in imidazole in FeMoco).

Of concern is the high oxidation state (V) in the model compounds compared to FeMoco (III) and the Mo-Mo bond (I'm assuming) in the dinuclear compounds that is less similar to FeMoco.

Re: Change as suggested. We have tried *R*-homocitrate and sundry homologues coupling with cysteine, histidine or imidazole to coordinate with molybdenum(IV/V/VI) salts under disparate synthesis conditions. The obtained complexes are precipitated or deprotonated products. The protonated structures **1** ~ **3** were synthesized for the first time by the two-step hydrothermal method. We have added the comparisons of Mo–O_{α-hydroxy/α-alkoxy}, Mo–O_{α-carboxy} and C–O_{α-hydroxy/α-alkoxy} bond distances (Å) for **1** ~ **3** with related molybdenum(IV/V/VI) lactates/glycolates complexes that have been reported. It is noted that protonation has a greater effect on bond lengths than metal valence or ligand types. Comparative results have been supplemented in Tables S10 and S11.

Also it is not clear how relevant the protonation state of extracted FeMoco is to the protein-bound FeMoco making the study less relevant for the enzyme chemistry. The structure of extracted cofactor is not completely known, there has been no crystal structure as far as I am aware. It is obvious that the cysteine and histidine ligands are no longer present, plausibly being replaced by NMF solvent ligands but it is not clear whether there are other structural changes in going from protein-bound cofactor to cofactor NMF solution.

Re: A direct comparison of the VCD spectra of the NMF-extracted FeMo-co (**20**) and other ≡C–O molybdenum α-hydroxycarboxylates suggest that *R*-homocitrate in FeMo-cofactor of molybdenum nitrogenase is protonated. Actually, the extraction method is based on literature reports with NMF to replace cysteine and histidine coordination. The extraction process has no influence with the redox state of FeMo-co and will not affect its recombination activity. Related description has been added in

the section of FeMo-co extraction from *Av1*.

The vibrational assignments made are also not entirely convincing. Is there no structure available for compound **13**? Since extracted FeMoco is being compared to this compound that seems like a crucial omission.

Re: Change as suggested. Compound **13** has been renamed as **21**, whose synthesis method is referenced in document,¹ only *R*-homocitrate is used for synthesis.

I would also strongly suggest the authors to collaborate with a computational chemist to put these band assignments on firmer ground.

Re: Change as suggested. We have conducted the density functional theory (DFT) calculations for molybdenum lactates **1** and NMF-extracted FeMo-co **20**. Detailed results and discussion have been added in Figures 5, 6 and pages 11 ~ 12 in manuscript.

Overall, while the work is interesting, the assignments are not entirely convincing and as this is extracted cofactor (without a completely known structure) means that the study is less relevant for nitrogenase and should rather be published in a more specialized journal.

Re: Change as suggested. We have added theoretical calculations for peak identification, which is suitable for the assignment of protonated *R*-homocitrate of FeMo-cofactor in molybdenum nitrogenase.

Other comments:

- The story is not well presented and there are inconsistencies in the text, e.g.

Line 199-200: “When homocitrate undergoes deprotonation, C–OH vibration will shift from 1084 to 1068 cm⁻¹”, shouldn’t this statement be the other way around?

Re: Change as suggested. This statement has been substituted with “When homocitrate undergoes protonation, C–OH vibration will shift from high wave number to low wave number, which is consistent with the red-shift observed for protonated *R*-lactate described above” on page 10.

- There are some language issues that should be taken care of.

- Line 30: The abbreviations used in the chemical formula of FeMoco seem inconsistent.

Re: Change as suggested. All abbreviations used in the chemical formula of FeMoco has been unified as FeMo-co.

- Line 199-200: I don’t think it is fair to say that when homocitrate is deprotonated, it shifts from 1068 cm⁻¹ to 1084 cm⁻¹ as these are two different compounds that are being compared.

Re: Change as suggested. This statement has been modified with “When homocitrate undergoes protonation, C–OH vibration will shift from high wave number to low wave number, which is consistent with the red-shift observed for protonated *R*-lactate

described above” on page 10.

- Figure 6: The peaks are labelled with bold numbers (1,2,3,4,5,6,7,8,9,10,11,12,13) which is the same as used when discussing the compounds. It would make the discussion clearer to use different labels.

Re: Change as suggested. The peaks are labelled with bold lowercase (a ~ e) in Figure 2. Original mark (1, 2, 3, 4, 5, 6, 7, 8, 9, 10, 11, 12, 13) in Figure 6 has been replaced by peak value directly in Figure 4.

Reviewer #2: In the present work, Zhou and co-workers synthesized and characterized two dinuclear (lactate) Mo(V) oxo/sulfido complexes **1** and **2** and a glycolate complex **3**. They measured the VCD spectra of **1**, **2**, some known Mo(VI)-oxo complexes, and extracted FeMo-cofactor, to analyze the possible protonation on the alpha-hydroxy group of *R*-homocitrate in the extracted FeMoco. In particular, the C–O/C–OH stretching frequencies were compared among the Mo complexes and extracted FeMoco, and the authors concluded the presence of a proton on the alpha-oxygen atom. The importance of the presence of a proton on the chelating alpha-hydroxy group of *R*-homocitrate (in the extracted FeMoco) has not been sufficiently discussed. Therefore, the importance of the work in the community remains unclear.

Re: Change as suggested. The structure of Mo-nitrogenase’ active site FeMo-cofactor has been clarified as Δ -Mo*Fe₇S₉C(cys)(His)[*R*-(H)homocit*] in Figure 1, where the local structure of the coordinated homocitrate has been temporarily suggested as tetravalent with deprotonated $\equiv\text{C}-\text{O}$ group or trivalent anion as Mo*Fe₇S₉C(cys)(His)(*R*-Hhomocit*) with protonated $\equiv\text{C}-\text{OH}$ group under physiological condition of nitrogenase. We have also added the description for the importance of the presence of proton “The protonation step confers a certain degree of lability to the homocitrate ligation to the cofactor,²⁻⁵ and thus allows structural flexibility that is important for the mechanism of N₂ reduction. Recent capture of N₂ species in FeMo-protein confirmed proton translocation and facilitate a serial of hydrogenation like α -hydroxy group.”⁶ in page 3.

What the authors mainly carried out in this work were synthesis, characterization, and spectroscopic analysis of a few Mo(V) oxo/sulfido complexes, which were unfortunately unable to assign the entitled “protonated *R*-homocitrate in FeMo-cofactor”. Therefore the title is misleading, in my opinion.

Re: Change as suggested. The title have been modified as “Assignment of protonated *R*-homocitrate in extracted FeMo-cofactor of nitrogenase *via* vibrational circular dichroism and DFT calculations” and the main work is spectroscopic analysis of model compound and extracted FeMo-co. Syntheses and characterization for model compounds have been transferred to supplementary materials.

The observed C–O frequency of 1068 cm⁻¹ for the extracted FeMo-cofactor is in between the C–O/C–OH vibrations of lactate complexes. As such, the authors cannot conclude the C–OH moiety in the FeMoco based on the new compounds. The authors

finally proposed the C–OH moiety in the extracted FeMoco, based on the comparison with a known (*R*-homocitrate) Mo(VI) oxo complex, which exhibits the C–O band at 1084 cm⁻¹. Thus, the conclusive evidence is not from the main body of the present work.

Re: Change as suggested. The present work is set up as labels for the further identification of the protonation state for *R*-homocitrate. When *R*-lactate undergoes protonated coordination, $\nu(\text{C–O})$ peak will move to low wave number. The same rule should be followed when the homolog becomes *R*-homocitrate. Complexes with same ligands are compared to reduce feasible errors. Actually, the known Mo(VI) oxo complex can be serve a new compound based on the different chirality of homocitrate reference to previous literature.

The number of closely relevant Mo compounds exceeds 50 for Mo(V) or Mo(VI) oxo or oxo/sulfido complexes, in which Mo atoms are supported by an N-donor ligand and a chelating carboxylate/alcohol or alkoxide. Therefore, synthesis and characterization of the present Mo complexes do not stand out.

Re: In fact, the protonated structures **1** ~ **3** were synthesized for the first time by the two-step hydrothermal method. Only a few molybdenum α -hydroxycarboxylates were reported with coordination of α -hydroxy (protonated) and α -carboxy groups like (Et₄N)₃[(Hcit)Mo(CO)₃], (Et₄N)₂[(Hmal)Mo(CO)₃], (Et₄N)₂[(Hmal)Mo(CO)₃], [Mo₃S₄(PPh₃)₃(Hlact)₂(lact)], K₂{Mo₃^{IV}O₄(im)₃[Mo^{VI}O₃(Hcit)]₂}·3im·4H₂O and (Him)₂{Mo₃^{IV}SO₃(im)₃[Mo^{VI}O₃(Hcit)]₂}·im·6H₂O.⁷⁻⁹ From these structures, we can see that most of them are citric acid which is achiral. Only [Mo₃S₄(PPh₃)₃(Hlact)₂(lact)] is mixed with protonated and deprotonated groups. Therefore, protonated molybdenum lactates **1** ~ **2** in this paper are representative.

With these reasons, I am sorry to conclude that the present work does not warrant sufficient novelty required for Communications Chemistry.

Additional comments:

a) Comparison of the C–O/C–OH frequencies of chelating lactate was made by compounds in different oxidation states, Mo(V) for C–OH and Mo(VI) for C–O. The authors need to clarify the effect of oxidation states before attributing the bathochromic shift to protonation/deprotonation behaviors. The Mo–O bond strength varies by the oxidation state, and the Mo–O bond strength should affect the neighboring C–O bond strength.

Re: Change as suggested. We have added the comparison of selected bond distances (Å) for **1** ~ **3** with related different oxidation states of molybdenum(IV/V/VI) lactates/glycolates complexes, which have been reported previously. The results illustrate that protonation has a greater effect on bond lengths than metal valence. Comparative results and description have been supplemented in Tables S10 and S11.

b) Along these lines, the oxidation states of “model” compounds are different from the extracted FeMoco. The Mo(V) and Mo(VI) states are irrelevant to the Mo atom of

FeMo-cofactor, which has been assigned as the Mo(III) state (in some cases proposed as the Mo(IV) state).

Re: Change as suggested. The Mo(III) state is much unstable. By analyzing Tables S10 and S11, we can see that the valence state, coordination modes of ligands have effects on the bond distances. But protonation has the stronger effect. In the new manuscript, we have changed to suggest that *R*-homocitrate is protonated in FeMo-cofactor in molybdenum nitrogenase.

Reviewer #3: The manuscript by Deng et al. describes a study that examines the protonation state of the homocitrate ligand of the FeMo-cofactor through a comparative spectroscopic analysis of the extracted FeMo-cofactor of Mo-nitrogenase with four organo-molybdenum compounds. While the available crystal structures of MoFe protein suggest that homocitrate is coordinated to the Mo atom of the FeMo-cofactor through the O atoms of its carboxyl (COO⁻) and alkoxy (CO⁻) groups, recent theoretical studies implied that the alkoxy ligand might be protonated into a COH group (C2 atom). Based on these calculations, it has been proposed that this protonation step confers a certain degree of lability to the homocitrate ligation to the cofactor, and thus allows structural flexibility that is important for the mechanism of N₂ reduction. The current study takes advantage of the chirality of the homocitrate COH group (C2 atom), which can be probed by vibrational circular dichroism (VCD) spectroscopy. The authors first characterized two Mo complexes coordinated by the *R*- and *S*-enantiomers of protonated-lactates (Hlac) and identified the COH vibrations around the 1050-1060 cm⁻¹ region through the mirrored VCD spectra of the two enantiomers. They then studied two Mo complexes coordinated by *R*- and *S*-lactates (lac) and identified their CO⁻ vibrations to be at around 1070-1090 cm⁻¹. Taken together, these results established that the protonated ligand had a CO(H) vibration at lower wavenumbers compared to its non-protonated counterparts. Based on these results, the authors then studied the NMF-extracted FeMo-cofactor using the same VCD technique and identified a peak at 1068 cm⁻¹ that was consistent with a vibration at 1084 cm⁻¹ and observed in the VCD spectra of an unprotonated *R*-homocitrate ligand of the Mo compound. This observation led the authors to conclude that the homocitrate of the FeMo-cofactor is ligated to Mo through a protonated COH group.

This study represents the first direct experimental attempt to prove the existence of a protonated ligation of homocitrate to the Mo atom of the FeMo-cofactor, which is of potential relevance to the mechanism of nitrogenase. However, despite its scientific appeal, the manuscript suffers from two major issues. One, the justification for the assignment of the VCD peaks to certain vibrations is unclear. For example, the authors based the crucial assignment of the peak at 1068 cm⁻¹ in the spectrum of FeMo-cofactor on the similarity of this peak to those observed in the spectra of the Mo complexes. The basis for such an assignment is not apparent to the non-experts. In addition, the VCD spectra presented by the authors are rather complex convoluted, and the interpretation of these spectra lacks any validation by results derived from complementary methods, such as DFT simulations and/or isotopic labeling experiments.

Re: Change as suggested. We have conducted the DFT calculations for molybdenum lactates **1** and the NMF-extracted FeMo-co **20**. Detailed results and discussion have been added in Figures 5, 6 and pages 11 ~ 12 in new manuscript.

Two, the relevance of the extracted FeMo-cofactor to its native protein bound counterpart is unclear. Given the fragile nature of the extracted FeMo-cofactor, it is questionable that the cofactor would remain intact after the extensive (8 hr) drying procedure. Moreover, even if the homocitrate ligand of the FeMo-cofactor indeed undergoes protonation, it is possible that this event occurred during the extraction procedure. Concerns along this line are not addressed by experiments or discussed in the manuscript and prevent publication of this manuscript in the current state.

Re: Change as suggested. The extraction has no influence with the redox state of FeMo-co and will not affect its recombination activity. Related description has been added in the section of FeMo-co extraction from *Av1*. The structure of the protein is complicated, we do not rule out possibility of protonation occurred during the extraction procedure. We have identified the NMF-extracted FeMo-co is protonated in new manuscript and further suggest that the *R*-homocitrate in iron-molybdenum prosthetic group of molybdenum nitrogenase is protonated.

There are also some minor issues that should be addressed by the authors:

1. Page 3, line 27: “their cofactor” should be “their cofactors”.

Re: It's regret that we did not find “their cofactor” in page 3, line 27.

2. The overlay of the two moieties in Figure 2a is hard to visualize. It would be helpful to use different levels of transparency or color schemes for each moiety.

Re: Change as suggested. The Figure 2a has been modified to Figure 1a and the overlay of the two moieties have been used with different levels of transparency.

3. The NMR data presented in the manuscript are not necessarily supportive for the conclusion of this study. These data should be moved into the Supplemental Materials.

Re: Change as suggested. NMR spectra and data have been moved to Supplemental Materials.

4. Illustrations of all compounds used in this study should be included in the manuscript.

Re: Change as suggested. Illustrations of all compounds used in this study have be included in the new manuscript.

5. A discussion of the significance of the findings of this work for a better understanding of nitrogenase mechanism should be included in the Conclusion section.

Re: Change as suggested. Related descriptions have been added in the conclusion.

6. Many parts of the manuscript are difficult to read (e.g., line 30-31, line 78-81, and line 140-143, etc.). The manuscript would benefit from a thorough proofreading for improved clarity.

Re: Change as suggested. Major revisions have been finished in new manuscript.

References

1. Zhou Z. H., Hou S. Y., Cao Z. X., Tsai K. R., Chow Y. L. Syntheses, spectroscopies and structures of molybdenum(VI) complexes with homocitrate. *Inorg Chem* **45**, 8447–8451 (2006).
2. Thorhallsson A. T., Benediktsson B., Bjornsson R. A model for dinitrogen binding in the E4 state of nitrogenase. *Chem Sci* **10**, 11110–11124 (2019).
3. Siegbahn P. E. M. The mechanism for nitrogenase including all steps. *Phys Chem Chem Phys* **21**, 15747–15759 (2019).
4. Dance I. The pathway for serial proton supply to the active site of nitrogenase: enhanced density functional modeling of the Grotthuss mechanism. *Dalton Trans* **44**, 18167–18186 (2015).
5. Cao Z. X., Jin X., Zhou Z. H., Zhang Q. E. Protonation of metal-bound α -hydroxycarboxylate ligand and implication for the role of homocitrate in nitrogenase: computational study of the oxy-bidentate chelate ring opening. *Int J Quantum Chem* **106**, 2161–2168 (2006).
6. Kang W., Lee C. C., Jasniewski A. J., Ribbe M. W., Hu Y. L. Structural evidence for a dynamic metallocofactor during N_2 reduction by Mo-nitrogenase. *Science* **368**, 1381–1385 (2020).
7. Takuma M., Ohki Y., Tatsumi K. Molybdenum carbonyl complexes with citrate and its relevant carboxylates. *Organometallics* **24**, 1344–1347 (2005).
8. Sokolov M. N. et al. Complexes of $M_3S_4^{4+}$ (M=Mo, W) with chiral α -hydroxy and aminoacids: synthesis, structure and solution studies. *Inorg Chim Acta* **395**, 11–18 (2013).
9. Wang S. Y., Zhou Z. H. Molybdenum imidazole citrate and bipyridine homocitrate in different oxidation states – balance between coordinated α -hydroxy and α -alkoxy groups. *RSC Adv* **9**, 519–528 (2019).

Reviewers' comments:

Reviewer #1 (Remarks to the Author):

The authors have submitted a considerably improved manuscript. With accompanying DFT calculations the results are now on firmer ground. It seems my main concerns have been dealt with.

The manuscript can probably be published now, although I note that the language of the text could be improved.

Reviewer #2 (Remarks to the Author):

The authors are clearly not willing (or unable) to address the issues raised by me. All key issues, not only raised by me but also from others, are about the relevance of the present study to analyze the coordination mode of R-homocitrate in the nitrogenase FeMo-cofactor and the importance of the coordination mode of R-homocitrate regarding the enzymatic function. Only minor changes in descriptions have been made, without answering any critical questions and comments. The authors repeat "Change as suggested", but surprisingly, NONE of the important points has been addressed. I would like to support this field of works, but unfortunately the present form is not yet satisfactory for publication, as important discussions are missing or very unclear. The authors may need to admit that this work has less relevance to the FeMo-cofactor.

Reviewer #3 (Remarks to the Author):

The authors significantly improved the manuscript in this revision by (1) re-illustrating most of the figures, (2) clarifying many issues in the text, (3) addressing most of the minor concerns, and (4) adding DFT calculations to support their assignment of spectral features, which is the most important revision they made to the manuscript. The revised manuscript is now more readable and the result are presented in a more convincing manner as compared to before.

However, the relevance of the proposed protonation of homocitrate of the extracted FeMoco still needs to be addressed in a more convincing way. First, the authors should highlight that the extracted FeMoco can be used to reactivate apo-nitrogenase in the main text instead of burying this crucial information in the supplemental information. Second, the authors attempt to address the concern regarding possible protonation of the cofactor during the extraction process by simply stating that the "extraction process has nothing to do with the redox state" and including a citation in the supplemental material. This doesn't seem adequate since the concern is about the protonation state, and not the redox state of the extracted cofactor. Third, the authors may want to look into the literature to find support for homocitrate being protonated in its native environment. I would recommend acceptance of the manuscript if these issues can be addressed.

Response to reviewers' comments

Reviewers' comments:

Reviewer #1 (Remarks to the Author):

The authors have submitted a considerably improved manuscript. With accompanying DFT calculations the results are now on firmer ground. It seems my main concerns have been dealt with.

The manuscript can probably be published now, although I note that the language of the text could be improved.

Re: Change as suggested, the manuscript has been revised with highlights in the main text.

Reviewer #2: (Remarks to the Author):

The authors are clearly not willing (or unable) to address the issues raised by me. All key issues, not only raised by me but also from others, are about the relevance of the present study to analyze the coordination mode of R-homocitrate in the nitrogenase FeMo-cofactor and the importance of the coordination mode of R-homocitrate regarding the enzymatic function. Only minor changes in descriptions have been made, without answering any critical questions and comments. The authors repeat "Change as suggested", but surprisingly, NONE of the important points has been addressed. I would like to support this field of works, but unfortunately the present form is not yet satisfactory for publication, as important discussions are missing or very unclear. The authors may need to admit that this work has less relevance to the FeMo-cofactor.

Re: In fact, the chelated mode of *R*-homocitrate remains unchanged regarding the enzymatic function in view of the recent structures bound with CO and N₂ substrates.^{1,2} The references have been added to the main text. Moreover, the extracted FeMo-co can be used to reactivate *apo*-nitrogenase, and will not affect its recombination activity.³⁻⁵ This supported that extraction process has nothing to do with the redox state including protonation state of FeMo-co.⁶ FeMo-co is also active for CO catalytic reduction to hydrocarbons like CH₄, C₂H₄, C₂H₆ without protein scaffolds.⁷⁻⁹ The related descriptions and the references have been added in discussion and the section of Extraction of FeMo-co from *Av1* and activity assays.

References

1. Spatzal T. et al. Ligand binding to the FeMo-cofactor: structures of CO-bound and reactivated nitrogenase. *Science* **345**, 1620–1623 (2014).
2. Kang W. et al. Structural evidence for a dynamic metallocofactor during N₂ reduction by Mo-nitrogenase. *Science* **368**, 1381–1385 (2020).

3. Shah V. K. & Brill W. J. Isolation of an iron-molybdenum cofactor from nitrogenase. *Proc. Natl. Acad. Sci. USA* **74**, 3249–3253 (1977).
4. Burgess B. K. The iron-molybdenum cofactor of nitrogenase. *Chem. Rev.* **90**, 1377–1406 (1990).
5. Fay A. W. et al. Characterization of isolated nitrogenase FeVco. *J. Am. Chem. Soc.* **132**, 12612–12618 (2010).
6. Huang H. Q., Kofford M., Simpson F. B. & Watt G. D. Purification, composition, charge, and molecular weight of the FeMo cofactor from *azotobacter vinelandii* nitrogenase. *J. Inorg. Biochem.* **52**, 59–75 (1993).
7. Lee C. C., Hu Y. L. & Ribbe M. W. ATP-independent formation of hydrocarbons catalyzed by isolated nitrogenase cofactors. *Angew. Chem. Int. Edit.* **51**, 1947–1949 (2012).
8. Lee C. C., Hu Y. L. & Ribbe M. W. Catalytic reduction of CN^- , CO, and CO_2 by nitrogenase cofactors in lanthanide-driven reactions. *Angew. Chem. Int. Edit.* **54**, 1219–1222 (2015).
9. Lee C. C. et al. A comparative analysis of the CO-reducing activities of MoFe proteins containing Mo- and V-nitrogenase cofactors. *ChemBioChem* **19**, 649–653 (2018).

Reviewer #3: (Remarks to the Author):

The authors significantly improved the manuscript in this revision by (1) re-illustrating most of the figures, (2) clarifying many issues in the text, (3) addressing most of the minor concerns, and (4) adding DFT calculations to support their assignment of spectral features, which is the most important revision they made to the manuscript. The revised manuscript is now more readable and the results are presented in a more convincing manner as compared to before.

However, the relevance of the proposed protonation of homocitrate of the extracted FeMoco still needs to be addressed in a more convincing way. First, the authors should highlight that the extracted FeMoco can be used to reactivate apo-nitrogenase in the main text instead of burying this crucial information in the supplemental information.

Re: Change as suggested, “cell growth and purification of nitrogenase proteins, extraction of FeMo-co from *Av1*” sections have been moved to the main text. “The extracted FeMo-co can be used to reactivate *apo*-nitrogenase” and the related descriptions have been added in Extraction of FeMo-co from *Av1* and activity assays section.

Second, the authors attempt to address the concern regarding possible protonation of the cofactor during the extraction process by simply stating that the “extraction process has nothing to do with the redox state” and including a citation in the supplemental material. This doesn’t seem adequate since the concern is about the protonation state, and not the redox state of the extracted cofactor.

Re: Change as suggested. In fact, the extraction process will not affect the recombination ability of FeMo-co. “The extracted FeMo-co can be used to reactivate *apo*-nitrogenase” has been added. The original sentence “extraction process has

nothing to do with the redox state” has been replaced by “The extraction process has nothing to do with the redox state including protonation state of FeMo-co and will not affect its recombination activity”.

Third, the authors may want to look into the literature to find support for homocitrate being protonated in its native environment. I would recommend acceptance of the manuscript if these issues can be addressed.

Re: Actually, we have been tracking the literatures on nitrogenase. Most of studies are about reduction mechanism of nitrogen fixation to ammonia and the structure simulation of catalytic active center FeMo-co. Because the exact protonated crystal structure has not been obtained until now, most of explorations remain in theoretical calculations. As far as we know, the experimental studies on protonated *R*-homocitrate is limited to our previous infrared spectra comparison,¹⁰ besides the proposed structure and the computational simulations. Direct experimental evidence is more difficult in view of protons are bitty and almost always invisible in limited-resolutions of crystal structures. In addition, we have added several references on the theoretical studies of the protonation of *R*-homocitrate in the introduction section.

In fact, the chelated mode of *R*-homocitrate remains unchanged regarding the enzymatic function in view of the recent structures bound with CO and N₂ substrates.^{1,2} The references have been added to the main text. Moreover, the extracted FeMo-co can be used to reactivate *apo*-nitrogenase, and will not affect its recombination activity.³⁻⁵ This supported that extraction process has nothing to do with the redox state including protonation state of FeMo-co.⁶ FeMo-co is also active for CO catalytic reduction to hydrocarbons like CH₄, C₂H₄, C₂H₆ without protein scaffolds.⁷⁻⁹

Namely,

1. The chelated mode of *R*-homocitrate remains unchanged even nitrogenase bound with CO and N₂ substrates respectively;
2. The extracted FeMo-co can be used to reactivate *apo*-nitrogenase;
3. Hydrocarbon formation like CH₄, C₂H₄, C₂H₆ by solvent-extracted cofactors proved CO can be reduced by cofactors without the assistance of corresponding protein scaffolds.

The related descriptions and the references have been added in discussion and the section of Extraction of FeMo-co from *Av1* and activity assays.

References

1. Spatzal T. et al. Ligand binding to the FeMo-cofactor: structures of CO-bound and reactivated nitrogenase. *Science* **345**, 1620–1623 (2014).
2. Kang W. et al. Structural evidence for a dynamic metallocofactor during N₂ reduction by Mo-nitrogenase. *Science* **368**, 1381–1385 (2020).
3. Shah V. K. & Brill W. J. Isolation of an iron-molybdenum cofactor from nitrogenase. *Proc. Natl. Acad. Sci. USA* **74**, 3249–3253 (1977).

4. Burgess B. K. The iron-molybdenum cofactor of nitrogenase. *Chem. Rev.* **90**, 1377–1406 (1990).
5. Fay A. W. et al. Characterization of isolated nitrogenase FeVco. *J. Am. Chem. Soc.* **132**, 12612–12618 (2010).
6. Huang H. Q., Kofford M., Simpson F. B. & Watt G. D. Purification, composition, charge, and molecular weight of the FeMo cofactor from *azotobacter vinelandii* nitrogenase. *J. Inorg. Biochem.* **52**, 59–75 (1993).
7. Lee C. C., Hu Y. L. & Ribbe M. W. ATP-independent formation of hydrocarbons catalyzed by isolated nitrogenase cofactors. *Angew. Chem. Int. Edit.* **51**, 1947–1949 (2012).
8. Lee C. C., Hu Y. L. & Ribbe M. W. Catalytic reduction of CN^- , CO, and CO_2 by nitrogenase cofactors in lanthanide-driven reactions. *Angew. Chem. Int. Edit.* **54**, 1219–1222 (2015).
9. Lee C. C. et al. A comparative analysis of the CO-reducing activities of MoFe proteins containing Mo- and V-nitrogenase cofactors. *ChemBioChem* **19**, 649–653 (2018).
10. Jin W. T. et al. Preliminary assignment of protonated and deprotonated homocitrates in extracted FeMo-cofactors by comparisons with molybdenum(IV) lactates and oxidovanadium glycolates. *Inorg. Chem.* **58**, 2523–2532 (2019).

REVIEWERS' COMMENTS:

Reviewer #3 (Remarks to the Author):

The authors have addressed all my concerns, and the manuscript is now ready for publication.